# Soft Bimodal Sensor Array Based on Conductive Hydrogel for Driving Status Monitoring

**DOI:** 10.3390/s20061641

**Published:** 2020-03-15

**Authors:** Wentao Dong, Daojin Yao, Lin Yang

**Affiliations:** 1School of Electrical and Automation Engineering, East China Jiaotong University, Nanchang 330013, China; wentao_dong@163.com (W.D.); ydaojin@whu.edu.cn (D.Y.); 2Department of Mechanical and Electrical Engineering, Huazhong Agricultural University, Wuhan 430070, China

**Keywords:** soft bimodal sensor array (SBSA), electronic skin (E-skin), conductive hydrogel, driver’s status monitoring

## Abstract

Driving status monitoring is important to safety driving which could be adopted to improve driving behaviors through hand gesture detection by wearable electronics. The soft bimodal sensor array (SBSA) composed of strain sensor array based on ionic conductive hydrogels and capacitive pressure sensor array based on ionic hydrogel electrodes is designed to monitor drivers’ hand gesture. SBSA is fabricated and assembled by the stretchable functional and structural materials through a sol–gel process for guaranteeing the overall softness of SBSA. The piezoresistive strain and capacitive pressure sensing abilities of SBSA are evaluated by the data acquisition system and signal analyzer with the external physical stimuli. The gauge factor (GF) of the strain sensor is 1.638 under stretched format, and –0.726 under compressed format; sensitivity of the pressure sensor is 0.267 kPa^−1^ below 3.45 and 0.0757 kPa^−1^ in the range of 3.45–12 kPa, which are sensitive enough to hand gesture detection and driving status monitoring. The simple recognition method for the driver’s status behavior is proposed to identify the driver’s behaviors with the piezoresistive properties of conductive polymers, and the turning angles are computed by the strain and pressure values from SBSA. This work demonstrates an effective approach to integrate SBSA seamlessly into an existing driving environment for driving status monitoring, expanding the applications of SBSA in wearable electronics.

## 1. Introduction

Nowadays, safety driving problems have received increased attention, in which a large number of crashes happen every day, threatening the lives and properties of human beings [1,2]. Most parts of the traffic accidents are related with inappropriate driving behaviors [3]. Hand gestures recorded by electronic sensors are strongly related to the driving behaviors for bridging the electronic system with safety driving behaviors which would promote prosthesis control [4], gesture recognition [5,6], and driving training [7]. Therefore, how to reduce traffic accidents would be studied in safety driving with sensing data analysis and fusion algorithms [8]. A variety of sensors, such as digital image sensors, accelerometers, and infrared sensors are developed to increase awareness and promote safety during the driving process [2,9,10]. Recently, epidermal electronics or soft electronic patches have been developed to monitor driving activities [7,11]. Testing and characterization of electronic sensors are used to improve the drivers’ ability [12]. Soft electronic skin (E-skin) is highly desired to distinguish the turning directions, driving status monitoring, and turning angels effectively. 

Flexible/stretchable electronics are designed and fabricated to the strain sensor for human gesture detection, driver status monitoring, and human robot interaction applications [13,14]. E-skin based on a liquid metal strain sensor is fabricated to realize the multisite sensing capabilities for touch ability feedback [15]. Human machine interface (HMI) based on strain sensor and electromyogram (EMG) data is designed to control the motion of quadrotor by different hand gestures [16]. The wearable porous pressure sensitive rubber (PPSR) pressure sensors and strain gauges are integrated into the smart glove to control the robot motion wirelessly [17,18]. Over the past years, electronic devices have been developed for driving status monitoring, but the lack of stretchability and lateral tensile sensing property was limited in body applications [19,20]. E-skin with high sensitivity is designed to driving status monitoring which would provide an effective way to hand gesture and driving behavior monitoring for safety driving.

Soft electronics are designed to biological health monitoring which contacts with the skin surface conformally for improving signal quality [21]. Traditional electrophysiological electrodes and sensors are used to physiological signal monitoring which would lead to discomfort to the drivers as incompatible properties at the skin/rigid sensor interface [1,22]. Soft materials with high senility and stretchability have been used to E-skin for hand gesture detection, motion detection, and human machine interaction [23,24]. The ionic conductive gel, as a typical soft material, is sensitive to external stimuli signal, which can meet the compatibility of the skin in mechanical and electrical properties [25,26,27,28]. Conductive ionic gels stand as alternative functional materials for stretchable devices, and have demonstrated successful applications in E-skins [29]. Resistance change of ionic gels is much lower than traditional electronic conductors under large strains [30]. Capacitive sensing pressure sensors with electric double layers based on ionic gels are designed for hand gesture detection [31]. Moreover, capacitive sensors for tactile sensing of E-skin have demonstrated high strain sensitivity, compatibility with static force measurement [32]. E-skin with excellent flexibility and stretchability was designed for personal conditions monitoring with continuously tracking physiological signals, such as, heart rate, breath and skin temperature, which was associated with body motion and driver’s status [33,34]. Flexible microneedle patches contacted with the complex topography of skin surface conformally are fabricated for human health and well-being measuring directly on the human body [35]. Soft electronics based on lab-on-skin with similar physical properties of human skin are implemented for continuous hand gesture monitoring and robotic feedback control [36]. However, human motions and gestures could be recognized with complex data fusion algorithms for driving status monitoring. E-skin should be designed to recognize the driving behavior easily using the piezoresistive properties of conductive polymers. The positive and negative values from the strain sensor based on piezoresistive conductive polymers would be adopted to distinguish turning directions, and E-skin integrated with strain and pressure sensor based on conductive polymers is designed to identify the driver’s behaviors.

The research purpose of this work is to introduce a novel bimodal sensor array composed of strain and pressure sensors based on the ionic conductive gel. The highly stretchable and transparent hydrogel is prepared through a sol–gel process. SBSA is designed with a strain and pressure sensing ability to ensure stable, sensitive, and long-term turning direction recognition and angle measurement. The main contribution of this work is concluded as following:
(1)Bimodal design and easy fabrication method of SBSA. SBSA is designed with a strain sensor based on ionic conductive gels and the capacitive pressure is based on ionic gel electrodes. The highly stretchable and transparent hydrogel is prepared through a low-cost, simple, and easy-fabricated sol–gel process. The sensor array is fabricated by the stretchable functional and structural materials to guarantee the overall softness of SBSA, which conformally contacts with the surface of the skin for stable, continuous, long-term drivers’ status monitoring.(2)Simple recognition method for the driver’s behaviors with the properties of conductive polymers. SBSA integrated with the strain and pressure sensor based on ionic gels is to identify the driver’s behaviors. A simple method is proposed to distinguish the turning right or left behaviors with negative and positive strain data from conductive ionic gels of SBSA. It would build up the relationship between the driving status and the strain/pressure data from SBSA. Combined with strain and pressure data values from SBSA, the responding bending angles could be computed to evaluate the driving behaviors. 

## 2. Materials Selection and Structural Design

### 2.1. Design Strategy

The structural design and principle of SBSA are illustrated in Figure 1, which is composed of strain sensor array based on ionic conductive gels and the capacitive pressure array based on ionic gel electrodes. The sensor array is fabricated and assembled by the stretchable functional and structural materials, which is used to guarantee the overall softness of SBSA. The schematic graph of SBSA with capacitive pressure and piezoresistive strain sensors is shown in Figure 1a. The pressure sensor unit is close to the strain sensor with a certain space in the horizontal direction. The whole bimodal sensor array is implemented with 3 × 2 sensor units, which would be adhered to one soft substrate. The sensor cell with strain and pressure sensor is depicted in Figure 1b. The sensing data from the strain and pressure sensors is recorded by the variable resistance/capacitance values due to the external physical stimuli, and the working principle of sensor array is shown in Figure 1c. The piezoresistive conductive ionic gel is designed to the strain sensor for collecting the variable output resistance value of the ionic gel under the external applied strain. The parallel plate capacitive pressure sensor is designed to record the variable pressure signal. The capacitance value of the pressure sensor with the conductive ionic gel electrodes is variable with the external stimuli. 

### 2.2. Fabrication

SBSA is composed of strain sensor array based on ionic conductive gels and the capacitive pressure array based on ionic gel electrodes. The conductive hydrogel, one typical soft material, is important to the strain and pressure sensing function of SBSA. The highly stretchable and transparent ionic conductive hydrogel is prepared through a sol–gel process [37,38], as shown in Figure 2a. The conductive gels are prepared by dissolving monomer powder and sodium chloride (NaCl) into deionized (DI) water. An amount of 15.62 g acrylamide (AAm) (Aladdin Co. China) power and 16.029 g NaCl are added into 100 mL DI water. Upon dissolving powder in DI water under agitation, 0.17 g ammonium persulfate (AP) monomer (Aladdin Co. China) is added into the prepared solution, followed by addition of 0.06 g chemical crosslinker N,N-methylenebisacrylamide (MBAA) (Aladdin Co. China). Afterward, 0.25 g N,N,N,N-tetramethylethylenediamine (TEMED) (Aladdin Co. China) are added to enhance the conductivity and increase the reaction rate. After crosslinking for 60 min, the monolithic hydrogel is peeled off from the petri dish. The gels are then immersed in an aqueous solution of the same concentration of NaCl solution for 24 h, which absorb more water and reach a new state of equilibrium. Figure 2b depicts the optical image of SBSA with capacitive pressure sensor and piezo-resitive strain sensor array. The unique binary networks composed of both chemically and physically crosslinked segments contribute to the high stretchability, as illustrated in Figure 2c. It is shown that the ionic gel could be conductive during the stretching and relaxing process as the crosslink in ionic solution would suffer larger deformation [39,40]. Figure 2d–f depicts the different deformation modes of the hydrogel–elastomer (i.e., stretching, compressing and twisting) and conformability with the complex surface of the soft object while maintaining its structural integrity. Figure 2d depicts transparence of SBSA with conductive ionic gel. Figure 2e,f depict the optical images of SBSA in the bended, stretched formats which indicate SBSA could suffer a larger deformation and be conformal in contact with skin surface. SBSA extends the potential application in the field of hand gesture monitoring and HMI with good compatibility with softness.

The strain and pressure sensors are adopted into SBSA based on the soft, transparent, conductive hydrogel. The piezoresitive property of the conductive hydrogel is designed to stretchable strain sensor through cutting the hydrogel into the certain size (4 mm × 1 mm), which would be assembled into the soft substrate for strain data recordings. The strain sensor with conductive ionic gels is used to monitor the deformation of the body for hand gesture recognition. The capacitive pressure sensor with conductive ionic gel electrodes is designed to record the tactile pressure data. The capacitive pressure sensor with VHB 4910 (3M Corp., US) as dielectric layer and conductive ionic gel as electrodes is designed to detect the pressure data. The size of the capacitive pressure (dielectric layer) is 2 mm × 2 mm. The size of bottom electrode of the capacitive pressure sensor is a little larger than the dielectric layer with 2.2 mm × 2.2 mm. The size of top electrode of the capacitive pressure sensor is a little smaller than the dielectric layer with 1.8 mm × 1.8 mm. The total size of the soft sensor array (soft PDMS substrate) is 40 mm × 60 mm. The pattern for carbon grease (847, MG Chemical Corp., Surrey, BC, Canada) lines is designed in the Auto CAD software. The screen template is fabricated with a designed pattern by one Cooperation in Suzhou, China. The carbon grease could be printed onto the soft sensor array via the manual screen printing machine where the designed pattern on the screen template would be aligned to soft sensor array to ensure that the wire is connected with both ends of the strain sensor and the top and bottom electrodes of the pressure sensor. Considering the low cost for manufacturing and easy access for measurement, screen printing technology is utilized to fabricate carbon grease as conductive electrodes for extracting the strain and pressure sensor data. The carbon grease is connected with the top/bottom electrodes of capacitive pressure sensor unit and two ends of the strain sensor unit.

## 3. Results and Application

### 3.1. Electrical Performance

Electrical stability is a vital property for strain/pressure data collection of SBSA with strain and pressure sensors as shown in Figure 3. The experiments for electrical tests and hand gesture detection of the soft sensor array are demonstrated around the room temperature 20 °C, humidity (~70%), and the electrical performance values (piezoresistive strain sensor and capacitive sensing pressure) of the sensor array would remain steady. The temperature/humidity effects for the strain and pressure sensor could be ignored during the experiment process as the stable testing environment. The piezo-resistive performance of hydrogel-based strain sensors is evaluated by the data acquisition system (NI PCI 6259, NI Corp., Austin, TX, USA) and the home-made stretcher controlled by PMAC motion controlling board. Figure 3a depicts that the variable resistance has the linear relationship with the external applied strain. It is represented as: ΔR/R_0_=1.64 × ε (R^2^ = 0.99) under the stretched format. It is also represented as: ΔR/R_0_=-0.726 × ε (R^2^ = 0.99) under compressed format, where ε is the external applied strain to the sensor, R_0_ is the initial value of the strain sensor without deformation, and ΔR is the variable resistance value under the external applied strain. Gauge factor (GF) is the paramount parameter for strain sensor which indicates the electrical response upon the strain and is defined as $GF=\frac{\Delta R/R_{0}}{\Delta \varepsilon }$. GF of the strain sensor in the stretched format (Figure 3a) is computed as 1.638, and −0.726 in the compressed format. Figure 3b depicts that the long-term monitoring of variable resistance value output under repeated loading and unloading process. It is validated that the stable resistance output (GF around 1.638 or −0.726) can be continuously read without obvious deviation, demonstrating the stability of the strain sensor. The stable properties of the strain sensor based on conductive ionic gel would provide an effective way to long-term hand gesture detection.

The capacitive pressure sensor performance with conductive hydrogel electrodes is evaluated by the data acquisition system and signal analyzer (Semiconductor character system, Keithley 4200-SCS, Keithley Corp., Solon, OH, USA) and the loading cell. With the increasing loading value to the capacitive pressure value, the output capacitance value also increases as shown in Figure 3c. Plum scattered dots mean measured data by experiments. Blue curve means connections for the experiment points. Red dotted line represents piecewise linear fitting curves for the experiment data, which is used to analyze the capacitive sensing pressure data. It is devoted that the variable capacitance value has an approximate linear relationship with the loading pressure value. It is represented as:$C=f(P)P+C_{0}$. Sensitivity is the paramount parameter for a pressure sensor which indicates the electrical response upon pressure and is defined as $S=\frac{\Delta C/C_{0}}{\Delta P}$, where ΔC is the capacitance value variation under certain pressure, C_0_ is the initial capacitance value of the parallel capacitance with ionic gel electrodes, and P is the pressure loading. Piecewise linear curve is adopted to fit the relationship between the capacitance value C and the pressure P. The segmentation point is pressure P=3.45kPa. Below 3.45 kPa, the relationship between the capacitance value C and the pressure P is represented as: C=0.267 × P+6.72. At the span of the pressure from 3.45 to 12 kPa, the relationship is represented as: C=0.0757 × P + 7.379. The sensitivity of our hydrogel-based pressure sensor is 0.267 kPa^−1^ below 3.45 kPa (R^2^ = 0.95), and 0.0757 kPa^−1^ in the range from 3.45 to 12 kPa (R^2^ = 0.97). Figure 3d depicts that the long-term monitoring of variable capacitance value output under repeated loading and unloading process. It is validated that the stable capacitance output is continuously read without obvious deviation, demonstrating the stability of the stain and pressure sensors. The excellent reproducibility of resistance and capacitance variations of conductive ionic gel during loading and unloading cycles has strongly confirmed the electrical stability of SBSA. The strain and pressure sensing ability are integrated into SBSA for providing more accurate gesture data to improve the driving behaviors. 

### 3.2. Strain and Pressure Distribution

SBSA is used to mimic the functions of the human skin which has been demonstrated by the sensing of the output signals of the pressure and strain sensors. For E-skin applications, a sensor with large-area array structure is required to record more data from the subjects, instead of a single pixel sensor. The NI PCI-5259 multi-channel data collection board is used to record the multi-pixel signal data from SBSA, where the data would be processed and analysis on the MATLAB software for pressure and strain data visualization. 

Figure 4a shows that the optical image of SBSA is bended to the certain state, and the corresponding strain and pressure data are plotted in Figure 4b,c, respectively. The bending radius curvature of the sensor pixel in the middle part is larger than the other pixel, and the corresponding strain and pressure data from the middle pixel (strain sensor: #3 and #4; pressure sensor: #c and #d) are larger than the other pixels as shown in Figure 4b,c.

Figure 4d depicts the optical image of SBSA when a wood is pressed onto SBSA (strain sensor: #1, #3, #5; and pressure sensor: #a, #c, #e). The pressure and strain data distribution from SBSA with the 3 × 2 pixels are distributed in Figure 4e–f. Both the strain and pressure data from the pixels of the wood sliding over the sensor units are larger than the other pixels. The strain and pressure data from other pixels that the wood does not touch over are nearly 0. The bimodal sensors are sensitive to the strain and pressure during wood pressed onto SBSA.

The real-time measurement of pressure and strain data with the 3 × 2 pixels bimodal sensor is demonstrated by touching the sensor by a pen as shown in Figure 4g. When a pen pressed onto the pixels of SBSA (strain sensor: #1, #2; and pressure sensor: #a, #b), the corresponding strain and pressure distribution are depicted in Figure 4h,i. The strain and pressure data from the pixels of the pen are larger than the other pixels.

### 3.3. Bending Angle Measurement

A mechanism of wrist joint with three degrees of freedom (DOF) is introduced to complicated actions, including flexion and extension, pronation and supination, radial and ulnar deviation [41]. SBSA is laminated onto the wrist for hand gesture monitoring (different bending angles) as shown in Figure 5a, which SBSA is used to collect the strain and pressure sensor data. The holding positions would play an important part for the driving behavior monitoring. Strain sensor units #3, #4, and pressure sensor units #c, #d of the sensor array are laminated onto the large deformation part of the wrist. Figure 5a depicts the optical images of SBSA laminated onto the back surface of wrist in different bending angles from −90 to 90° with 30° interval. It is defined as the minus angle when the right wrist bends outside (the first three frames from left to right in Figure 5a) where SBSA is in the compressed format. The output strain data value is negative when the driver’s right wrist turns the steering wheel right. The positive angles are defined as the right wrist bends inside (the last three frames from left to right in Figure 5a) where SBSA is in the stretched format. The strain data from SBSA is also positive when the driver’s right wrist turns the steering wheel left. The positive and negative strain data from SBSA would be used to distinguish whether the hand is turning right or turning left, which is closely related with the driving status behaviors.

Figure 5b depicts the strain data distribution during the bending process with different bending angles. The stain data from the sensor units increase with the bending angles (from 0 to 90° or from 0 to −90°). The strain data from six strain sensor units is depicted in Figure 5b with different lines. It is found that positive strain data is for turning left and negative strain data is for turning right. The strain data from the sensor units (#3 and #4) is larger than other sensor units as the deformation of sensor #3 and #4 is larger than other parts. Figure 5c depicts the local enlarged graph of Figure 5b for illustrating the strain data distribution from other pixels of SBSA (strain sensor: #1, #2, #5, #6). Experiments have demonstrated that the strain and pressure data distribution from SBSA are related with the location of the sensor array as the variable deformability of the monitoring subjects.

Figure 5d depicts the pressure data distribution during the bending process with different bending angles. There are six pressure sensor units embedded into SBSA. The pressure data from the sensor units increase with the bending angles (from 0 to 90° or from 0 to −90°). The pressure values from SBSA with the turning right and left actions is both positive as the output of capacitance value of the capacitive sensitive pressure sensor is varied in the positive range. It is shown that the pressure data from sensor pixels (#c and #d) is also larger than other pixels as the external pressure applied onto the pixels are also larger during the bending process of the wrist. Figure 5e depicts the local enlarged graph of Figure 5d for better illustrating the pressure data distribution from other pixels of SBSA (pressure sensor: #a, #b, #e, #f). 

The output values of the strain sensor are positive or negative when strain sensor based on ionic conductive gel is in the stretched or compressed formats. When SBSA is laminated onto the wrist, these properties will provide an effective way to distinguish whether turning right or turning left. The turning angles are related with the concrete values of the strain and pressure data in the different bended states. The obvious feature from SBSA would provide potential application in human machine interaction, hand gesture monitoring, and drivers’ status monitoring.

### 3.4. Hand Gesture Monitoring

SBSA is applied to hand gesture detection successfully (Figure 5), which is related with the driving behavior such as manipulating the steering wheels (turn right or left with certain angles). The driving state monitoring experiment platform is built for capturing the motion data of the hands during operating the steering wheel, including SBSA, NI PCI-5259 multichannel data collection unit, industry controller, and MATLAB software. The turning angles of the steering wheel is controlled by the bending angles of the wrists, which could be measured by SBSA with the strain and pressure data distribution. The concrete bending angles of wrists during the manipulating steering wheel process are computed to predict the turning angles of the steering wheel. The strain and pressure data distribution of different hand gestures are recorded by SBSA under different turning angles of the steering wheel.

Figure 6a shows that the optical image of SBSA laminated onto the wrist for manipulating the steering wheels (turning right) to control the motion of the car with different bending angles. The down frame of the corresponding pressure and strain data distribution recorded by SBSA with the 30° turning angles are shown in Figure 6b–c. The resistance and capacitance values of the soft bimodal sensor array raise to different levels with an increase of its tensile deformation caused by the bending angles of the wrist during the driving process. Figure 6d–e depicts the strain and pressure data distribution from SBSA with the 60° turning angles for manipulating the steering wheel. The strain and pressure data distributions in the 90° turning angles for manipulating the steering wheel are depicted in Figure 6f–g. It is found that the negative output of the strain data distribution is captured for turning the steering wheel right actions. In addition, the output values of the strain data increase with the increasing turning angles. The strain data from #3 and #4 pixels are also larger than other pixels during the turning actions as the #3 and #4 pixels of SBSA is right above the bending wrist where the part suffers larger deformation. The pressure data is also increased with increasing turning angles during the driving process. The pressure data from #c and #d pixels are also larger than other pixels of SBSA. Combined with strain and pressure data values from bimodal sensor array, the responding bending angles would be also computed to evaluate the driving behaviors and improve the driving ability.

Figure 6h shows the optical images of SBSA laminated onto the wrist surface for manipulating the steering wheels (turning left) to control the motion of the car with different bending angles. The corresponding pressure and strain data distribution recorded by SBSA with the 30° turning angles is shown in Figure 6i,j. Figure 6k,l depicts the strain and pressure data distribution from SBSA with the 60° turning angles for manipulating the steering wheel. The strain and pressure data distributions in the 90° turning angles for manipulating the steering wheel are depicted in Figure 6m,n. It is found that the output values of the strain data increase with the increasing turning angles. The strain data from #3 and #4 pixels are also larger than other pixels during the turning actions as the #3 and #4 pixels of SBSA is right above the bending wrist where the part suffers larger deformation. The pressure data is also increased with increasing turning angles during the driving process.

It provides a simple method to distinguish whether turning right or left with negative and positive strain data from SBSA, the output strain value from the hydrogel strain sensor has great differences with obvious features. The negative strain output would be output as it is compressed, and the positive strain output by stretching the polymer strain sensor. It is devoted as the relationship between the driving status and the strain and pressure data from SBSA. The changing rules of strain sensor and capacitive pressure sensor data is concluded for driving behavior recognition. Combined with strain and pressure data values from SBSA, the responding bending angles would be also computed to evaluate the driving behaviors and improve the driving ability.

The soft bimodal sensor array (SBSA) integrated with strain, pressure, and other sensing abilities are applied to hand gesture detection, drivers’ status monitoring, and HMI applications. Several effects (sensing ability, stretchability, fabrication, and sensitivity) are adopted to evaluate the performances of SBSA for hand gesture detection and drivers’ status monitoring as shown in Table 1. E-skin with pressure/temperature stimuli exhibited high pressure sensitivity of 0.7 kPa^−1^ up to 25 kPa and reproducible temperature coefficient of resistance of 0.83% K^−1^ in the temperature range 22–70 °C [42]. Flexible bimodal sensor using a paper platform and inkjet printing method exhibits a high-pressure sensitivity and high-endurance characteristics. [43]. A self-powered smart safety belt is developed to monitor the forward position and turning actions of the driver with high sensitivity (0.89 V/cm^2^) for strain measurements in the large strain domain (40–100%) [7]. This manuscript illustrates that SBSA composed of strain and pressure sensor array based on conductive ionic hydrogels is designed to detect drivers’ hand gesture. The simple recognition method for the driver’s status behavior is proposed to identify the driver’s behaviors with the piezoresistive properties of conductive polymers (positive value in the stretched format; negative value in the compressed format). Combined with strain and pressure data values, the responding bending angles would be also computed to evaluate the driving behaviors and improve the driving ability.

## 4. Discussion and Conclusions

The soft bimodal sensor array (SBSA) integrated with a strain and pressure sensor based on transparent ionic conductive gels has been designed and applied to hand gesture detection and drivers’ status monitoring. Therefore, the sensing ability, electrical performance, and data process method of SBSA would be promoted in several aspects for hand gesture detection. (1) Intelligent sensing and data fusion algorithms are needed to promote the recognition of driving status. The corresponding strain and pressure data could be recorded for extracting the typical features. SBSA could be also applied to different wrist gestures recognition with strain and pressure data distribution and intelligent data process technology. (2) Integration of SBSA into the steering wheels for helping the disabled with more sensory and intelligent manipulation ability. SBSA would be used as a human machine interface for the drivers to interact with outside environments or cars. (3) SBSA are laminated onto the two wrists for hand gesture detections. Combined with the sensing data from the two wrists, the hand gestures could be detected for helping driving status monitoring via data fusion algorithms. 

SBSA composed of strain and pressure sensor based on the soft functional and structural materials, is fabricated and assembled by ionic conductive gels for guaranteeing the overall softness. The capacitive pressure and strain sensing function of SBSA is evaluated by data acquisition system and the external physical stimuli. GF of the strain sensor 1.638 under stretched format, and −0.726 under compressed format; sensitivity of the pressure sensor is 0.267 kPa^−1^ below 3.45 kPa and 0.0757 kPa^−1^ in the range from 3.45 to 12 kPa, which is sensitive enough for hand gesture detection. SBSA laminated onto the wrist surface is applied to the driving state monitoring successfully (turning right, turning left) by the strain and pressure distribution, which the positive and negative values from the strain sensor array are used to distinguish turning right or turning left behaviors. The turning angles are computed by the concrete pressure and strain values from SBSA which is applied to the safety driving areas for driving status monitoring. This work demonstrates an effective approach to integrate SBSA seamlessly into the existing driving environment for driving status monitoring, and expanding the applications of SBSA in wearable electronics.

## Figures and Tables

**Figure 1 sensors-20-01641-f001:**
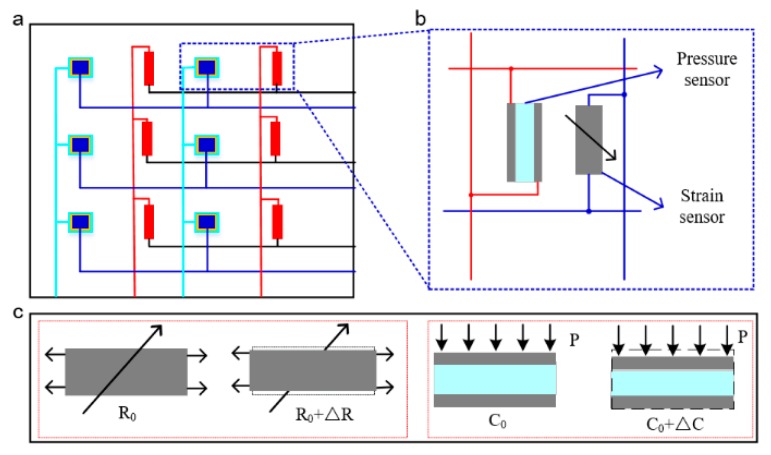
Structural design and principle of the soft bimodal sensor array (SBSA). (**a**) Schematic graph of SBSA; (**b**) sensor unit with pressure and strain sensors for SBSA; (**c**) principle of the strain and pressure sensor.

**Figure 2 sensors-20-01641-f002:**
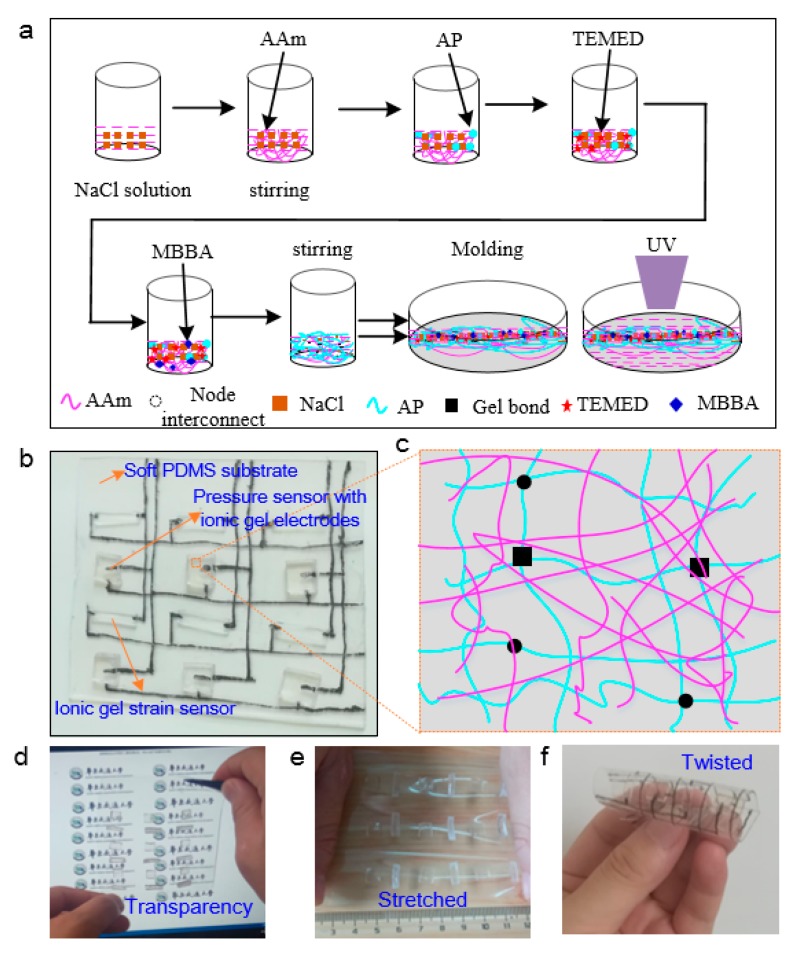
Fabrication process and integration of conductive ionic gel. (**a**) Sol–gel process for conductive ionic hydrogel fabrication; (**b**) optical image of transparent conductive ionic hydrogel; (**c**) schematic graph of the transparent ionic hydrogel; (**d**) transparence of SBSA; (**e**,**f**) optical images of SBSA in the stretched and bended formats.

**Figure 3 sensors-20-01641-f003:**
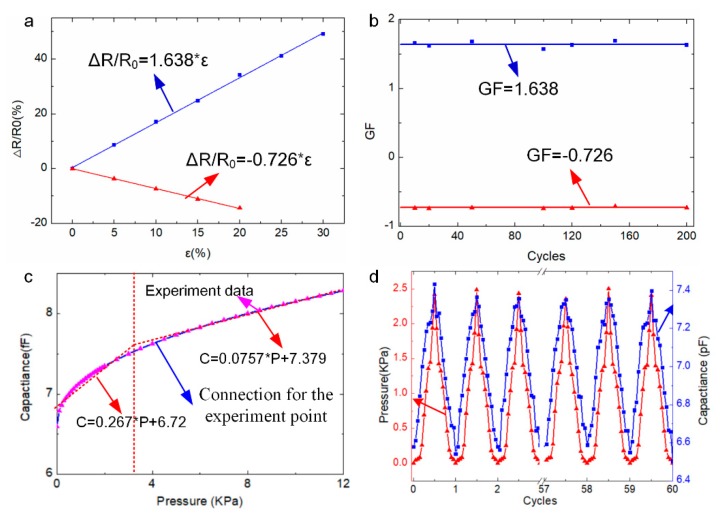
Electrical performance of SBSA. (**a**) Variable resistance value versus the strain data in the stretched and compressed process; (**b**) repeated strain data with different loading cycles; (**c**) relationship between the capacitance and external pressure; (**d**) repeated capacitance data with repeated loading cycles.

**Figure 4 sensors-20-01641-f004:**
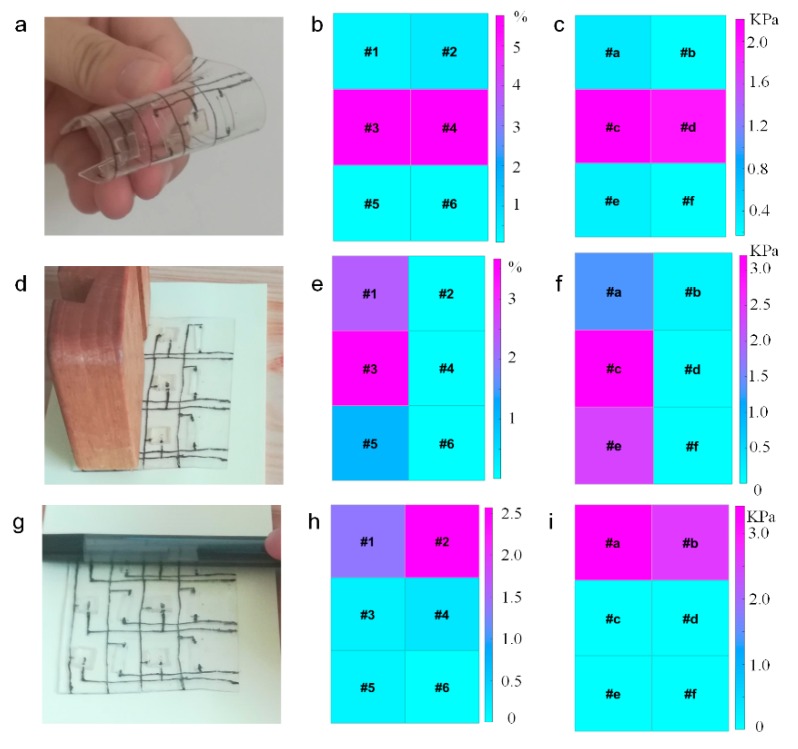
Pressure and strain data visualization based on SBSA. (**a**) Optical image of SBSA in the bended format; (**b**) strain and (**c**) pressure distribution of the bimodal sensor in the bended format; (**d**) optical image of a wood pressed on the SBSA; (**e**) strain and (**f**) pressure distribution during the wood pressed on the sensor array; (**g**) optical image of a pen pressing on SBSA; (**h**) strain and (**i**) pressure data from SBSA during the pen pressing on the sensor array.

**Figure 5 sensors-20-01641-f005:**
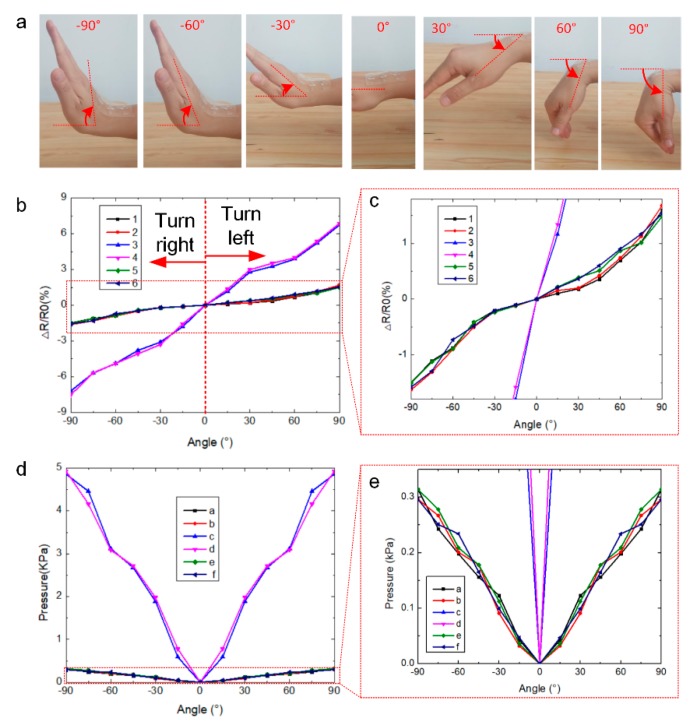
Pressure and strain data visualization based on SBSA. (**a**) Optical images of SBSA; (**b**) strain and (**d**) pressure data distribution of the bimodal sensor in the bended formats ranging from −90 to 90°. The corresponding local enlarged (**c**) strain and (**e**) pressure data of (**b**) and (**d**).

**Figure 6 sensors-20-01641-f006:**
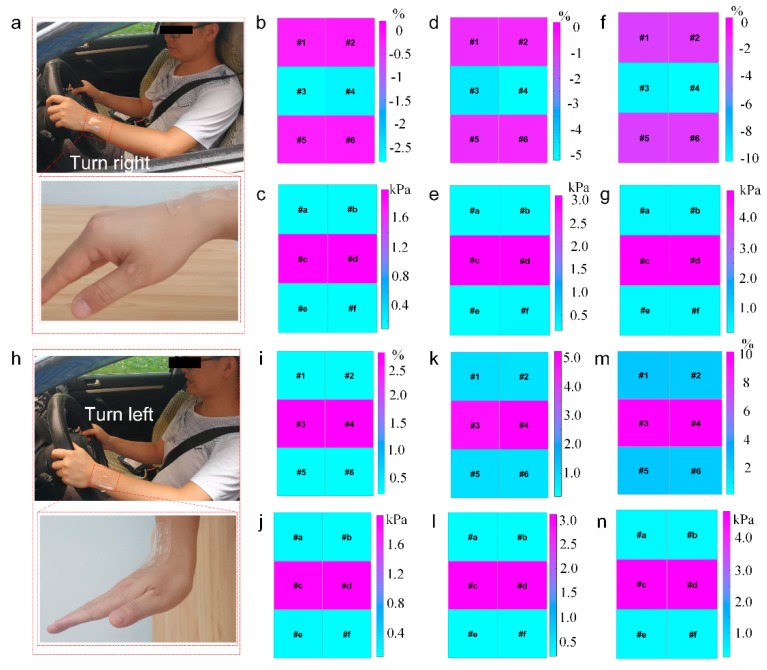
Driving state recognition via SBSA. (**a**) Optical images of SBSA laminated onto the wrist surface to manipulate the steering wheels for turning the car right; (**b**), (**d**), (**f**) strain and (**c**), (**e**), (**g**) pressure distribution recorded by SBSA with turning right the steering wheel 30, 60, 90°, respectively. (**h**) Optical images of SBSA laminated onto the wrist surface to manipulate the steering wheels for turning the car left; (**i**), (**k**), (**m**) strain and (**j**), (**l**), (**n**) pressure distribution recorded by SBSA with turning left the steering wheel 30, 60, 90°, respectively.

**Table 1 sensors-20-01641-t001:** Performances of SBSA.

Sensor Array	Sensing Ability	Stretch Ability	Fabrication	Sensitivity
Literature [41]	Pressure/temperature	<2%	MEMS	0.7 kPa^−1^
Literature [42]	Pressure/temperature	<3%	Inject print	--
Literature [7]	Strain	>20%	MEMS	0.89 V/cm^2^
This manuscript	Strain/pressure	>15%	sol–gel process	1.638/0.267 kPa^−1^

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
