# Peer review of "Soft Bimodal Sensor Array Based on Conductive Hydrogel for Driving Status Monitoring"

_sensors, 2020, doi:10.3390/s20061641_

Round 1

Reviewer 1 Report

This work proposed a soft bimodal sensor array (SBSA) which integrated with strain and pressure sensor based on transparent ionic conductive gels. The capacitive pressure and strain sensing function of SBSA is evaluated by data acquisition system and the external physical stimuli and driver’s behaviors and turning angles could be identified. The SBSA with good compatibility and softness shows a good application in driving status monitoring. Therefore, I recommend accepting the paper for publication after major revision:
1) The curves in Figure 3c have different colors of blue, plum, and red. What do the colors mean? I did not find any captions or legends for this.
2) There are some errors in the comment of Figure 4. Figure 4d depicts the optical image of SBSA when a wood is pressed onto SBSA but in your comment, it is the pen pressed on it.
In page 7, “Fig.5d depicts the strain data distribution during the bending process …” the “strain data distribution” should be “pressure data distribution”.
3) In my opinion, we can monitor hand gesture through strain and pressure data. In section 3.3, is the strain data distribution and pressure data distribution related to the position where SBSA is laminated onto the wrist? If the factor has obvious impact, could you give some clear conditions about the experiment? And is the data related to hand size of different people?
4) In section 3.4, how does it affect when holding different position of the steering wheel?
5) In section 3.4, does SBSA laminated onto the one or two wrists? How to monitor driving status with data from both wrists? What’s the relationship between the two sets of data?

Author Response

Replies to the Reviewer #1

Comment 1:The curves in Figure 3c have different colors of blue, plum, and red. What do the colors mean? I did not find any captions or legends for this.

Reply: Thanks to the reviewer for pointing out the deficiencies.

Plum scattered dots mean the measured data by experiments with certain pressures applied to the capacitive sensor. Blue curve means connection for the experiment points. Red dotted line represents piecewise linear fitting curves for the experiment data, which is used to analyze the capacitive sensing pressure data. It has been corrected highlighted in green font in Lines 181-184, Page 5.

Arrows in the red and blue colors has also been added in Figure 3d for detailed representations. And Figure 3 has been updated in the manuscript in Page 7.

Comment 2:There are some errors in the comment of Figure 4. Figure 4d depicts the optical image of SBSA when a wood is pressed onto SBSA but in your comment, it is the pen pressed on it. 

In page 7, “Fig.5d depicts the strain data distribution during the bending process …” the “strain data distribution” should be “pressure data distribution”. 

Reply: Thanks to the reviewer for pointing out the representation errors.

The comment of Figure 4 has been corrected highlighted in green font in Line 226 Pages 6-7.

“Fig.5d depicts the strain data distribution during the bending process …” has been corrected into “Fig.5d depicts the pressure data distribution during the bending process …” highlighted in green font in Line 258 Page 7.

Comment 3:In my opinion, we can monitor hand gesture through strain and pressure data. In section 3.3, is the strain data distribution and pressure data distribution related to the position where SBSA is laminated onto the wrist? If the factor has obvious impact, could you give some clear conditions about the experiment? And is the data related to hand size of different people?

Reply: Thanks to the reviewer for proposing the interesting problems.

The strain and pressure data distributions are related to the position where SBSA is laminated onto the wrist. In the experiments for hand gesture detection and driver’s behavior monitoring, strain sensor units #3, #4, and pressure sensor units #c, #d of the sensor array are laminated onto the large deformation part of the wrist which would provide more accurate strain and pressure sensing data. In my opinion, the sensing data from the sensor array is also related to hand size of different people. We can design several types of the sensor arrays with different sizes (very large, large, middle and small).

Experiment conditions are important to SBSA, and the electrical performances of SBSA would be changed with different experiment conditions, such as, temperature, humidity. The experiments for electrical tests and hand gesture detection of SBSA are demonstrated around the room temperature 20℃, humidity (~70%), and the electrical values (piezoresistive strain sensor and capacitive sensing pressure) of the sensor array would keep steady. The temperature/ humidity effects for the strain and pressure sensor could be ignored during the experiment process as the stable testing environment. It has been revised highlighted in yellow font in Line 171, Page 5.

Comment 4:In section 3.4, how does it affect when holding different position of the steering wheel?

Reply: Thanks to the reviewer for proposing the interesting problems.

SBSA would deform when hand bending and maniplating the steering wheel. The strain and pressure data would be recorded during the process. If holding different position of the steering wheel, the strain and pressure sensor units in SBSA would output different sensing values. The holding positions would play important part for the driving behavior monitoring. In the manuscript, strain sensor units #3, #4, and pressure sensor units #c, #d of the sensor array are laminated onto the large deformation part of the wrist. And the drivers also hold the left (right) end of the steering wheel which is parallel to car head. It has been revised in Discussion Section highlighted in green font in Line 253, Page 7.

Comment 5:In section 3.4, does SBSA laminated onto the one or two wrists? How to monitor driving status with data from both wrists? What’s the relationship between the two sets of data?

Reply: Thanks to the reviewer for proposing the interesting problems.

In the manuscript, SBSA is laminated onto the one wrist for hand gesture monitoring. The output strain data value is negative when driver’s right wrist turns the steering wheel right. The strain data from SBSA is also positive when driver’s right wrist turns the steering wheel left. The positive and negative strain data from SBSA would be used to distinguish whether hand turning right or turning left, which is closely related with the driving status behaviors.

If SBSA are laminated onto the two wrists, they could be also used to hand gesture detections. Strain and pressure data would be collected by SBSA from the two wrists during the driving process. If turning right, the strain sensor from left wrist is stretched, and the strain sensor from right wrist is compressed. The positive strain data is from left wrist, and negative value from right value. The turning direction could also been recognized. If turning left, the opposite sensing values could be detected by sensor array, respectively. The turning angles are also related with the concrete values from sensing data value from SBSA during driving process. It has been revised in Discussion Section highlighted in green font in Line 373, Page 11.

Reviewer 2 Report

In line 112, I think that the correct idea should be: The parallel plate capacitive pressure sensor is designed to record the variable pressure signal. Because, in the document the word "force" is mentioned only once.

In order to understand the section " Design strategy", it is neccesary to include the dimensions corresponding to each sensor. 

As it can see in figure 2, in section "fabrication", the explanation about fabrication process can be improved in order to explain the patterning process of each sensor and interconection lines. 

In section "Electrical Performance", the values of resistance, gauge factor and capacitance are based on geometrical parameters; however, the characteristics about each sensor like, thickness, dimensions, permittivity and other are not mentioned in the article.

Author Response

Replies to the Reviewer #2

Comment 1:In line 112, I think that the correct idea should be: The parallel plate capacitive pressure sensor is designed to record the variable pressure signal. Because, in the document the word "force" is mentioned only once.

Reply:Thanks to the reviewer for point out the mistakes.

It has been corrected into “The parallel plate capacitive pressure sensor is designed to record the variable pressure signal” highlighted in yellow font in Line 112, Page 3.

Comment 2:In order to understand the section "Design strategy", it is necessary to include the dimensions corresponding to each sensor. 

Reply:Thanks to the reviewer for pointing out the deficiencies.

The pressure sensor unit is close to the strain sensor with a certain space in the horizontal direction. The thickness of the fabricated conductive hydrogel film is about 0.5mm. The size of strain sensor unit with conductive gels is 4mm*1mm, and the size of the capacitive pressure (dielectric layer) is 2mm*2mm. The size of bottom electrode of the capacitive pressure sensor is a little larger than the dielectric layer with 2.2mm*2.2mm. The size of top electrode of the capacitive pressure sensor is a little smaller than the dielectric layer with 1.8mm*1.8mm. The total size of SBSA (the soft PDMS substrate) is 40mm*60mm. The corresponding representations are revised highlighted in yellow font in Line 150, Page 4.

Comment 3:As it can see in figure 2, in section "fabrication", the explanation about fabrication process can be improved in order to explain the patterning process of each sensor and interconnection lines. 

Reply:Thanks to the reviewer for pointing out the deficiencies.

The conductive hydrogel, one typical soft material, is important to strain and pressure sensing function of SBSA. The highly stretchable and transparent ionic conductive hydrogel is prepared through a sol–gel process. The conductive hydrogel film is cut into certain shape for strain and pressure assembly. The carbon grease (847, MG Chemical Corp., Canada) as conductive line is printed onto the two ends of the strain sensor for extracting the strain sensing data.

The capacitive pressure sensor with VHB 4910 (3M Corp., US) as dielectric layer and conductive ionic gel as electrodes is designed to detect the pressure data. Screen printing technology is utilized to fabricate carbon grease as conductive lines for extracting the pressure sensor data, which is connected with the top/bottom electrodes of capacitive pressure sensor unit.

The pattern for carbon grease lines is designed in the Auto CAD software. Screen template is fabricated with designed pattern by one Cooperation in Suzhou, China. The carbon grease could be printed onto SBSA via the manual screen printing machine where the designed pattern on the screen template would be aligned to SBSA to ensure that the wire is connected with both ends of the strain sensor and the top and bottom electrodes of the pressure sensor. The corresponding representations are revised highlighted in yellow font in Line 155, Page 4.

Comment 4:In section "Electrical Performance", the values of resistance, gauge factor and capacitance are based on geometrical parameters; however, the characteristics about each sensor like, thickness, dimensions, permittivity and other are not mentioned in the article.

Reply:Thanks to the reviewer for pointing out the deficiencies.

The size of strain sensor unit with conductive gels is 4mm*1mm, and the size of the capacitive pressure (dielectric layer) is 2mm*2mm. The thickness of the conductive hydrogel film is about 0.5mm. The corresponding representations are revised highlighted in yellow font in Line 154 Page 4.

The experiments for electrical tests and hand gesture detection of SBSA are demonstrated around the room temperature 20℃, humidity (~70%), and the electrical performance values (piezoresistive strain sensor and capacitive sensing pressure) of the sensor array would keep steady. The temperature/ humidity effects for the strain and pressure sensor could be ignored during the experiment process as the stable testing environment. The corresponding representations are revised highlighted in yellow font in Line 168, Page 5.

Reviewer 3 Report

The name of the article is very ambiguous. In this study it is not measured the tension and pressure of a human subject, but the position of some parts of the human body is measured using sensors which operate on the principle of measuring the strain and pressure. Therefore, you must change the title of the paper.

Cap. 2

Sensor design and construction are very interesting. The question is whether this process has repeatability. How many sensors were produced and how many were tested? What was the testing methodology? What was the calibration methodology? Have the results been similar?

Have you tried a comparison with sensors already on the market that are already verified?

Cap.3.2.

Under what conditions do the sensors retain their characteristics? Does it work well at any temperature, humidity or pressure? Have you tested these things?

Cap. 3.3

The procedure for measuring the wrist angles seems fine. How many people have been tested? Could you make a database with the angular values ​​of the wrist movement? Could you draw a conclusion on hand movements?

What guarantee do you have that the sensor is attached on the hand in the correct position. How do you determine the optimal position of attachment to the human body?

Cap.3.4

How many people have you tested? How does the position of the hand in the driving process differ from person to person?

The steering wheel rotation is a personalized movement from one man to another, which I do not think can be standardized.

How did you check the measured data with your sensor?

I propose to identify another human activity that involves hand movement, so that it can be controlled and classified.
The movement of the steering wheel is far too complex and customized to be able to generalize a method of tracking the movement.

Author Response

Replies to the Reviewer #3

Comment 1:The name of the article is very ambiguous. In this study it is not measured the tension and pressure of a human subject, but the position of some parts of the human body is measured using sensors which operate on the principle of measuring the strain and pressure. Therefore, you must change the title of the paper.

Reply:Thanks to the reviewer for pointing out the deficiencies.

The title “Soft bimodal sensor array for driving status monitoring through simultaneous strain and pressure detection” has been changed to “Soft bimodal sensor array based on conductive hydrogel for driving status monitoring”. It has been highlighted in blue font in page 1.

Comment 2:Sensor design and construction are very interesting. The question is whether this process has repeatability. How many sensors were produced and how many were tested? What was the testing methodology? What was the calibration methodology? Have the results been similar? Have you tried a comparison with sensors already on the market that are already verified?

Reply:Thanks to the reviewer for proposing the interesting problems.

3*2 strain and 3*2 pressure sensor units are produced and integrated into SBSA. For electrical performance testing, one strain sensor and one pressure are tested by external strain and pressure respectively. The piezo-resistive performance of hydrogel-based strain sensors is evaluated by data acquisition system (NI PCI 6259, NI Corp., US) and the home-made stretcher controlled by PMAC motion controlling board. The capacitive pressure sensor performance with conductive hydrogel electrodes is evaluated by data acquisition system and signal analyzer (Semiconductor character system, Keithley 4200-SCS, Keithley Corp., US) and the loading cell.

The results for strain sensors are similar during the repeated stretching and compressing process (Fig.3b), and the results for pressure sensors are similar during the repeated loading and unloading process.

We have compared performance SBSA with other sensors in the literatures, which have similar electrical performances. We have not verified the performance of the soft sensor on the market.

Comment 3:Under what conditions do the sensors retain their characteristics? Does it work well at any temperature, humidity or pressure? Have you tested these things?

 Reply: Thanks to the reviewer for proposing the interesting problems.

The experiments for electrical tests and hand gesture detection of SBSA are demonstrated around the room temperature 20℃, humidity (~70%), and the electrical performance values (piezoresistive strain sensor and capacitive sensing pressure) of the sensor array would keep steady. The temperature/ humidity effects for the strain and pressure sensor could be ignored during the experiment process as the stable testing environment.

We do not test the environment effect for SBSA, and we just do the experiments of SBSA in the environment with room temperature 20℃, humidity (~70%). The corresponding representations are revised highlighted in yellow font in Line 172, Page 5.

Comment 4:The procedure for measuring the wrist angles seems fine. How many people have been tested? Could you make a database with the angular values ​​of the wrist movement? Could you draw a conclusion on hand movements?

Reply:Thanks to the reviewer for proposing the interesting problems.

There are two people tested in the experiments for hand gesture detection and driving status monitoring.

To be honest, it is difficult for us to make a database with the angular values ​​of the wrist movement. The key of the manuscript is to design and fabricate SBSA for hand gesture detection, and the effectiveness of SBSA is validated for hand gesture detection. It is very interesting to make a database with angular values of the wrist movement during the driving process. We will make a database in the future research.

In my opinion, the hand movements could be concluded to the bending actions during the driving process for manipulating the steering wheel.

Comment 5:What guarantee do you have that the sensor is attached on the hand in the correct position. How do you determine the optimal position of attachment to the human body?

Reply:Thanks to the reviewer for proposing the interesting problems.

We have done some tests for SBSA which are laminated onto the different parts of wrist, and different sensor units are aminated onto the large deformation part of the wrist. In the end, we decide the strain sensor units #3, #4, and pressure sensor units #c, #d of the sensor array are laminated onto the large deformation part of the wrist.

SBSA would deform when hand bending and maniplating the steering wheel. The strain and pressure data would be recorded during the process. If holding different position of the steering wheel, the strain and pressure sensor units in SBSA would output different sensing values. The holding positions would play important part for the driving behavior monitoring. In the manuscript, strain sensor units #3, #4, and pressure sensor units #c, #d of the sensor array are laminated onto the large deformation part of the wrist. And the drivers also hold the left (right) end of the steering wheel which is parallel to car head. It has been revised in Discussion Section highlighted in green font in Line 252, Page 7.

Comment 6:How many people have you tested? How does the position of the hand in the driving process differ from person to person?

Reply:Thanks to the reviewer for proposing the interesting problems.

In our experiments, there are two people tested for hand gesture detection by SBSA. The sensor array is laminated onto wrist for hand gesture detection. As the position of the hand in the driving process differ from person to person, several types of the sensor arrays with different sizes (very large, large, middle and small) could be designed to hand gesture detection.

Comment 7:The steering wheel rotation is a personalized movement from one man to another, which I do not think can be standardized.

Reply:Thanks to the reviewer for proposing the interesting problems.

The steering wheel rotation is a personalized movement, and intelligent algorithms would be studied for extract the features during the driving process with the help of SBSA. Exploring intelligent algorithms and soft electronics into hand gesture detection and driving statue monitoring capabilities represents the future research direction.

Comment 8:How did you check the measured data with your sensor?

Reply:Thanks to the reviewer for proposing the interesting problems.

The piezo-resistive performance of hydrogel-based strain sensors is evaluated by data acquisition system (NI PCI 6259, NI Corp., US) and the home-made stretcher controlled by PMAC motion controlling board. The capacitive pressure sensor performance with conductive hydrogel electrodes is evaluated by data acquisition system and signal analyzer (Semiconductor character system, Keithley 4200-SCS, Keithley Corp., US) and the loading cell. Signal analyzer (Semiconductor character system, Keithley 4200-SCS, Keithley Corp., US) is measurement equipment with high test accuracy and stability, which could grantee the electrical tests of SBSA. The strain and pressure functionability of SBSA are calibrated by Semiconductor character system with external stimuli signals.

Round 2

Reviewer 1 Report

I recommend to accept in present form.

Author Response

Thanks to the reviewer for the hard work.

Reviewer 2 Report

Consider that this work represents a lot of time of dedication and experimentation.
The results obtained provide relevant information in the development of flexible electronics.
It is important that you consider that characterization of this SBSA sensors in a limited group of people
does not usually represent a reliable statistic, however, the development of this research is a substantial
contribution that in the future can be improved.
If principal objective of this technology is be integrated into vehicles for driving monitoring in a real situation,
before they should be evaluated with the different variables presents in that case.

Author Response

Replies to the Reviewer #2

Comment 1:It is important that you consider that characterization of this SBSA sensors in a limited group of people does not usually represent a reliable statistic, however, the development of this research is a substantial contribution that in the future can be improved.

Reply: Thanks to the reviewer for proposing the useful suggestions.

SBSA would be applied to collect strain and pressure data in a large group of people in the future. This manuscript mainly focuses on the effectiveness of hand gesture detection and drivers’ status monitoring by SBSA. The strain and pressure data from SBSA with different hand gestures in a large group of people would be trained and classified by AI technology for improving adaptability of SBSA in practical applications in the future.

Comment 2:If principal objective of this technology is be integrated into vehicles for driving monitoring in a real situation, before they should be evaluated with the different variables presents in that case. 

Reply: Thanks to the reviewer for proposing the interesting problems.

It is very difficult for SBSA to be integrated into vehicles for driving monitoring in the present time. It is demonstrated that SBSA is successfully applied to hand gesture detection and driving state monitoring with the help of strain and pressure data. But more work should be done before practical applications of SBSA for driving status monitoring. More stable sensor array should be designed and fabricated for continuous gesture data recordings. Data collection board and process unit should be developed to collect sensing data from SBSA. More people would be tested using SBSA for hand gesture detection and driving status monitoring, and intelligent algorithms would be studied to hand gesture detection and driving statue monitoring.

Reviewer 3 Report

I appreciate the idea of this sensor and the work done. However, I have some observations: 1. tests on human subjects are inconclusive. You only tested two people, the results cannot be generalized. 2. the example of using this sensor in the drivers hand is not appropriate. The movements of the driver's hands are endless in directions and values, it is very difficult to synthesize such a volume of movements only by measuring with this sensor. Therefore, I suggest finding another example, for which there is a limited range of movements that can be tracked and recorded. 3. The wrist is a very complex joint; it is not a simple rotation couple. I think there are many movements: flexion and extension, pronation and supination, radial and ulnar deviation. How can you track and record the movements of this joint with just this sensor? 4. you should make a kinematic scheme of the joint (based on the literature - mechanisms), to see clearly what the measured angles are. 5. I find it sufficient to test a few human subjects and, in parallel, to make the same movements on a simple device at which the angles can be measured very correctly.

Author Response

Replies to the Reviewer #3

Comment 1:Tests on human subjects are inconclusive. You only tested two people, the results cannot be generalized.

Reply: Thanks to the reviewer for proposing the interesting problems.

The strain sensor units #3, #4, and pressure sensor units #c, #d of the sensor array are laminated onto the large deformation part of the wrist, which could be used to detect the hand gestures (bending angles).

There are two people tested in the experiments for hand gesture detection and driving status monitoring, which the soft sensor array is applied to hand gesture detection and drivers’ status monitoring effectively. To be honest, the results could not be generalized. But several types of the sensor arrays with different sizes (very large, large, middle and small) could be designed to hand gesture detection for satisfying different human subjects. Exploring intelligent algorithms and soft electronics into hand gesture detection and driving statue monitoring capabilities represents the future research direction for extracting more general results. It has been revised in Discussion Section in blue font in Line 312, Page 9.

Comment 2:The example of using this sensor in the drivers hand is not appropriate. The movements of the driver's hands are endless in directions and values, it is very difficult to synthesize such a volume of movements only by measuring with this sensor. Therefore, I suggest finding another example, for which there is a limited range of movements that can be tracked and recorded. 

Reply: Thanks to the reviewer for proposing the interesting suggestions.

In my opinion, the drivers’ hands would move in a limited range during the manipulating the steering wheel process. Both the bending angles of the wrist and rotating angles of the steering wheel are limited. The bending angles of the wrist are successfully detected via the soft sensor array as shown in Fig.5. The drivers’ statuses are strongly related with the bending angles of the wrists when manipulating the steering wheel. Soft bimodal sensor array could be used to driving status monitoring effectively.

Comment 3:The wrist is a very complex joint; it is not a simple rotation couple. I think there are many movements: flexion and extension, pronation and supination, radial and ulnar deviation. How can you track and record the movements of this joint with just this sensor?

Reply: Thanks to the reviewer for proposing the interesting problems.

In the manuscript, the bending actions of wrist are monitored by the soft bimodal sensor array. The strain and pressure data could be collected by SBSA, and the strain data array is represented as data matrix: s=[s1, s2; s3, s4; s5, s6]; the pressure array represented as p=[p1, p2; p3, p4; p5, p6]. Fig.6 depicts the strain and pressure data distribution of different hand gestures are recorded by SBSA under different turning angles of the steering wheel. The bending angles and driving behaviors are detected by SBSA with concrete strain and pressure distribution value.

For other movements of the wrist joint, such as, flexion and extension, pronation and supination, radial and ulnar deviation, SBSA could also collection the corresponding strain and pressure data distribution which could be applied to analyze the wrist motion behaviors. When the wrist joint do flexion action, the SBSA would collect the sensing data s=[s1, s2; s3, s4; s5, s6] and p=[p1, p2; p3, p4; p5, p6]. Data features could be extracted with the flexion action combined with different strain and pressure units. If the wrist executes different actions, the corresponding strain and pressure data could be recorded for extracting the typical features. Intelligent algorithms would be studied to extract the features with the typical actions. It is shown that SBSA could be applied to different wrist gestures recognition with strain and pressure data distribution and intelligent data process technology. Complex joint behavior monitoring would be studied with SBSA in the further work. It has been revised in blue font in Line 367, Page 11.

Comment 4:You should make a kinematic scheme of the joint (based on the literature - mechanisms), to see clearly what the measured angles are.

Reply: Thanks to the reviewer for proposing the interesting suggestions.

A mechanism of wrist joint with three DOF is introduced to complicated actions [41]. The angle and stiffness control method of wrist joint are studied to improve motion accuracy. The motion of the joint is very complex, including flexion and extension, pronation and supination, radial and ulnar deviation. However, during the process of manipulating the steering wheel, the wrist would bend to a certain angle which would be detected by SBSA as shown in Fig.5. The drivers’ status are strongly related with the bending angles of the wrists. Soft bimodal sensor array could be used to driving status monitoring effectively. It has been revised in blue font in Line 250, Page 7.

References:

[41] K. Koganezawa and H. Yamashita, "Three DOF wrist joint - control of joint stiffness and angle," in 2010 IEEE International Symposium on Industrial Electronics, 2010, pp. 1973-1979.

Comment 5: I find it sufficient to test a few human subjects and, in parallel, to make the same movements on a simple device at which the angles can be measured very correctly.

Reply: Thanks to the reviewer for proposing the interesting problems.

Traditional simple devices could be used to measure joint angles correctly, but it would lead discomfort to the drivers as the incompatible properties at the skin/rigid sensor interface, especially in the long-term driving process. Soft bimodal sensor array based on ionic conductive gel is sensitive to external stimuli signal, which can meet the compatibility of the skin in mechanical and electrical properties. SBSA could follow the deformation of wrist, which is conformal contact with skin surface, which would provide more accurate strain and pressure data of hand gestures. It has been revised in blue font in Line 56, Page 2.